# Benchmarking the Reproducibility of Brain Tissue Segmentation Across MRI Scanners

**Ekaterina Kondrateva**[1,2] (iD)        EKATERINA.KONDRATEVA@MAASTRICHTUNIVERSITY.NL

**Abdalla Z Mohamed**[3] (iD)

**Sandzhi Barg**[4] (iD)

**Florian Kofler**[5,6,7,8] (iD)

[1] *Department of Radiation Oncology (Maastro), GROW Research Institute for Oncology and Reproduction, Maastricht University Medical Centre+, Maastricht University, Maastricht, The Netherlands*

[2] *Atelic AI, UAE*

[3] *Department of Cognitive Sciences, College of Humanities and Social Sciences, United Arab Emirates University, UAE*

[4] *Higher School of Economics (HSE), Russia*

[5] *Hertie Institute for AI in Brain Health, University of Tübingen, Tübingen, Germany*

[6] *Department of Quantitative Biomedicine, University of Zurich, Zurich, Switzerland*

[7] *Helmholtz AI, Helmholtz Munich, Neuherberg, Germany*

[8] *AI for Image-Guided Diagnosis and Therapy, TUM School of Medicine and Health, Technical University of Munich, Munich, Germany*

**Editors:** Accepted for publication at MIDL 2026

## Abstract

Accurate and reproducible brain morphometry from structural magnetic resonance imaging is critical for monitoring neuroanatomical changes across time and imaging domains. Although deep learning has accelerated segmentation workflows, scanner-induced variability and limited reproducibility remain major obstacles, particularly in longitudinal and multi-site studies. In this study, we benchmark two state-of-the-art segmentation pipelines, *FastSurfer* and *SynthSeg*, integrated into *FreeSurfer*, one of the most widely adopted neuroimaging tools. Using two complementary datasets—a 17-year single-subject longitudinal cohort and a nine-site test–retest cohort—we quantify between-scan segmentation variability with region-wise overlap and distance measures, including the Dice similarity coefficient, surface Dice, the 95th percentile of the Hausdorff distance, and the mean absolute percentage error in regional volumes.

Our results reveal up to 7–8% variation in the volumes of small subcortical structures such as the amygdala and ventral diencephalon, even under controlled test–retest conditions. This level of noise raises a critical question: can we reliably detect subtle longitudinal changes of 5–10% in small brain regions with volumes below 2 milliliters, given the magnitude of scanner- and site-induced morphometric variability? We further analyze how registration choices and interpolation modes contribute additional, although smaller, biases, and we show that surface-based quality filtering can remove outlier segmentations while preserving most scans and maintaining morphometric stability. This work provides a reproducible benchmark of modern FreeSurfer-based segmentation pipelines and highlights the need for harmonization and quality-control strategies to enable robust morphometry in real-world neuroimaging studies.

The code is available on [github.com/kondratevakate/brain-mri-segmentation](github.com/kondratevakate/brain-mri-segmentation)

**Keywords:** Machine Learning, Brain Morphometry, MRI, Multi-Scanner Variability, Dice, FreeSurfer, SynthSeg, Segmentation, Statistics, Test-Retest, Domain Shift

## 1. Introduction

Advances in AI-driven medical imaging have revolutionized pathology detection, yet reproducible morphometric analysis of healthy brains—especially across scanners and over time—remains a challenge. This gap limits our ability to monitor individual brain health trajectories and detect early pathological changes. While artificial intelligence (AI) has significantly advanced medical imaging, particularly in pathology segmentation tasks such as tumor identification in the BraTS challenge (Menze et al., 2015), there remains a notable gap in applying these advancements to morphometric analyses of healthy brains across varied domains. This underexplored area presents opportunities for developing robust, generalizable AI models that can accurately capture subtle anatomical variations, thereby deepening insight into brain aging and development.

Traditional tools like FreeSurfer (Fischl, 2012b) have been instrumental in providing detailed morphometric analyses of brain structures' volume, cortical surface area, and thickness. Recent integrations, such as SynthSeg (Billot et al., 2023c), offer contrast-agnostic segmentation capabilities trained on synthetic data, aiming to improve generalizability across different imaging protocols. In parallel, FastSurfer (Henschel et al., 2020b) emerged as a deep learning-based alternative to the traditional FreeSurfer pipeline, utilizing convolutional neural networks trained on real FreeSurfer-labeled data to achieve significantly faster processing times while maintaining comparable accuracy. Unlike SynthSeg's domain-randomization approach with synthetic training data, FastSurfer leverages supervised learning on anatomically labeled T1-weighted images, making it well-suited for standard clinical protocols but potentially less robust to imaging variations. Despite these advancements, challenges persist in ensuring reproducibility of volumetric estimates under real-world conditions, particularly when dealing with data from multiple scanners and protocols.

This study aims to assess the consistency of brain volume measurements using FastSurfer and FreeSurfer 8 with integrated SynthSeg across longitudinal MRI scans from a single individual. By quantifying inter-scan variability using metrics like absolute volume difference, Dice, and Surface Dice, we seek to highlight the limitations of current segmentation pipelines in personalized brain health monitoring and early detection of neurodegenerative conditions.

## 2. Related Works

Deep learning has significantly advanced individual-level brain morphometry from structural MRI. Traditional pipelines such as *FreeSurfer* (Fischl, 2012b) have long served as a gold standard, producing cortical and subcortical morphometric features (e.g., thickness, volume, surface area). However, these methods are computationally intensive and sensitive to scanner variability, limiting their scalability in large-scale or multisite studies.

Recent versions of FreeSurfer integrate *SynthSeg* (Billot et al., 2023c), a contrast-agnostic segmentation model trained on synthetic data. *SynthSeg* provides robust volumetric estimates across diverse contrasts, resolutions, and scanners. Its compatibility with

standard atlases (e.g., Desikan-Killiany, MUSE) makes it suitable for harmonized morphometry across heterogeneous datasets.

To address runtime bottlenecks, *FastSurfer* (Henschel et al., 2020b) provides a FreeSurfer-compatible alternative using a voxel-size independent convolutional neural network (FastSurferVINN) (Henschel et al., 2022), enabling accurate whole-brain segmentation and optional surface-based cortical analysis within minutes. Tools such as BrainChop (Tudosiu et al., 2023) prioritize clinical scalability, though often at the cost of generalization to unseen protocols.

Other high-performing segmentation models include *nnU-Net* (Isensee et al., 2021) and *nnFormer* ((Zhou et al., 2023)), which yield excellent accuracy in controlled benchmarks but often require dataset-specific finetuning to generalize effectively in clinical or real-world settings.

## 2.1. Longitudinal Modeling and Individualized Morphometry

The reproducibility of brain segmentation directly impacts downstream applications that rely on longitudinal morphometric stability, including disease progression modeling, normative deviation detection, and biological age estimation. Without consistent volumetric measurements across scans, even sophisticated longitudinal models risk misinterpreting noise as biological change, undermining clinical utility in personalized monitoring.

Normative modeling frameworks enable the estimation of z-score deviations from large-scale population references. This approach is particularly effective in identifying early deviations in psychiatric populations and supports both clinical and subclinical applications (Marquand et al., 2016).

Another widely adopted line of work focuses on brain age prediction. *BrainAGE* (Franke and Gaser, 2012) models estimate biological aging based on MRI-derived morphometric features, frequently using *FreeSurfer* outputs. These models have demonstrated strong longitudinal reliability and clinical interpretability.

Emerging tools like *Neurofind* (Vieira et al., 2025) offer user-friendly platforms that integrate normative modeling and brain age estimation, providing individualized reports based on high-resolution structural MRI images.

Despite these advances, challenges remain in achieving sulcal-level surface precision, quantifying uncertainty, and ensuring reproducibility in real-world multisite studies. Although morphometry has clear clinical applications, including epilepsy-focused MRI analysis and dementia-oriented volumetry (Aliev et al., 2021; Khadhraoui et al., 2024), rigorous longitudinal reproducibility benchmarks remain scarce.

## 2.2. Brain Morphometry as a Biomarker

Longitudinal MRI studies have greatly expanded our understanding of how brain morphometry changes over time, particularly in response to aging, disease, and stress. A growing body of work highlights structural biomarkers in specific brain regions—especially the hippocampus, anterior cingulate, and prefrontal cortex—that reflect vulnerability or resilience to neuropsychiatric conditions.

In healthy populations (Papagni et al., 2011) demonstrated gray matter volume (GMV) reductions in the anterior cingulate cortex (ACC), hippocampus, and medial prefrontal

cortex (mPFC) in individuals exposed to stress. Similar findings were confirmed in large-scale aging studies, including (Schaefer et al., 2018), who reported consistent hippocampal atrophy associated with aging. MacDonald and Pike (2021) provide a broader review of region-specific atrophy across the lifespan. Structural biomarkers also inform psychiatric research. Cardoner et al. (2024) review evidence of stress-induced degeneration in the ACC and dorsolateral prefrontal cortex (dlPFC), while Carnevali and Sgoifo (2018) identify preserved amygdala volumes as potential resilience markers. UK Biobank analyses further support longitudinal volume reductions in fronto-limbic circuits among individuals with high stress exposure (Statsenko et al., 2022). Importantly, several studies have examined structural changes within individuals undergoing therapy. Gryglewski et al. (2019) found hippocampal and amygdalar volume increases after electroconvulsive therapy (ECT) in treatment-resistant depression. Furtado et al. (2012) reported volumetric growth in the dlPFC after rTMS. Frodl et al. (2008) showed that psychotherapy attenuated gray matter loss over three years in depression. Together, these findings suggest that MRI-based brain morphometry, especially when assessed longitudinally, provides meaningful biomarkers for brain health across both normative and pathological aging.

## 3. Methods

We study the reproducibility of brain MRI segmentation pipelines across longitudinal and multi-site datasets. We employ two publicly available datasets: SIMON (Single Individual volunteer for Multiple Observations across Networks) and SRPBS (Strategic Research Program for Brain Sciences), spanning a wide range of scanners and protocols. We compare segmentation outputs from FreeSurfer, FastSurfer, and SynthSeg, using FreeSurfer's `recon-all` pipeline as a reference. Segmentation reproducibility is evaluated using a targeted subset of cortical and subcortical ROIs most relevant for neuroimaging biomarkers. For surface-based comparisons, we apply rigid registration using ANTs (Avants et al., 2011) and assess the effect of different interpolation modes and reference spaces. Quantitative evaluation is performed using Dice coefficient (DSC), Surface Dice, 95th percentile Hausdorff distance (HD95), and mean absolute percentage error (MAPE) of regional brain volumes. These metrics can be efficiently computed using tools such as panoptica (Kofler et al., 2024).

### 3.1. Datasets

**SIMON Dataset (Duchesne et al., 2019):** This dataset comprises 73 T1-weighted MRI scans of a single healthy male subject, collected over 17 years across multiple sites and 1.5T scanners (Duchesne et al., 2019), multiple sites with 35 distinct scanner settings.

**SRPBS Traveling Subject Dataset (Tanaka et al., 2021):** This dataset includes 411 T1-weighted MRI scans from 9 healthy subjects, each scanned at 9 different sites using 3T MRI scanners in constitutive days. The data is organized following the BIDS format and includes accompanying metadata such as participant demographics and scanner parameters (Tanaka et al., 2021).

Figure 1 provides a representative qualitative example from the SRPBS Traveling Subject dataset, illustrating scanner/protocol-induced domain shift at the image level and its downstream impact on segmentation consistency.

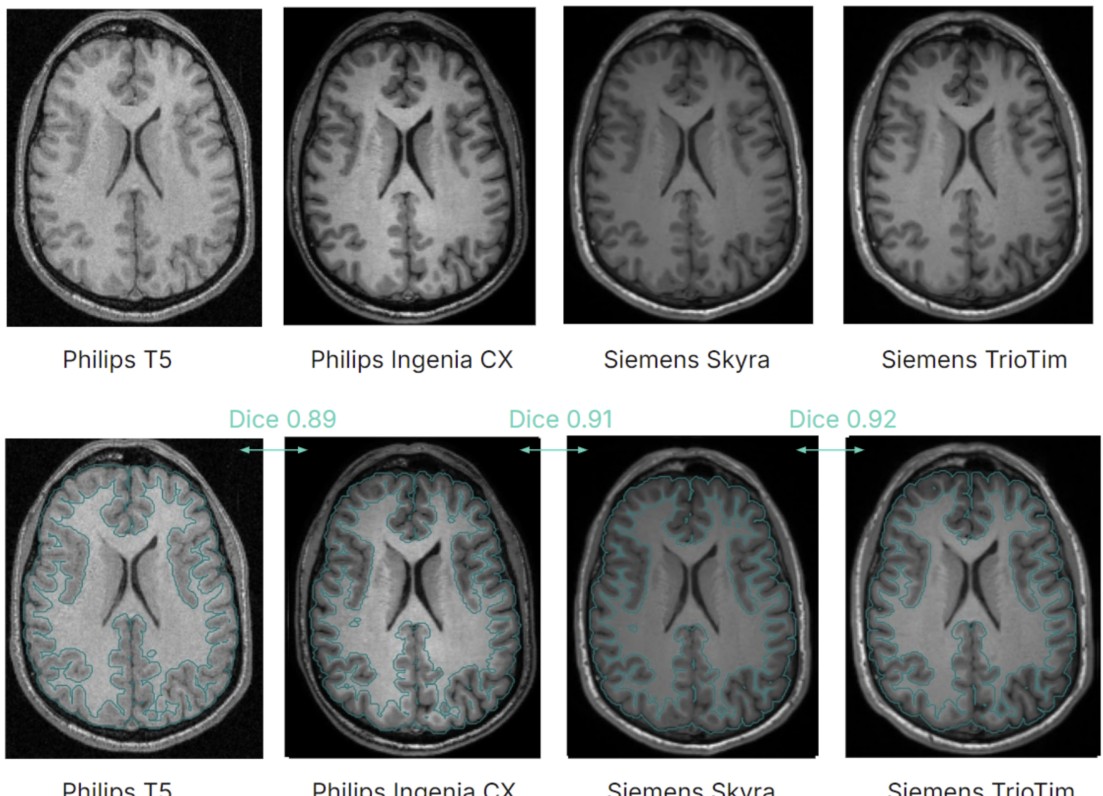

Figure 1: Representative qualitative example from the SRPBS Traveling Subject dataset (Subject 1; four sites/scanners; consecutive-day acquisitions). **Top:** T1-weighted scans of the same subject acquired on different scanners/sites, illustrating scanner/protocol-induced appearance differences (same anatomical slice; consistent display settings). All scans were rigidly registered to the first acquisition, and FreeSurfer `recon-all` label maps were co-registered accordingly; after `recon-all` conforming, images share the same voxel grid (size and spacing). **Bottom:** FreeSurfer outputs overlaid as contours derived from a binarized full-parcellation mask. Dice overlap values (range $[0, 1]$) are shown between adjacent acquisitions to summarize cross-scanner segmentation consistency. Even when anatomy is held constant and scans are aligned to a common space, scanner/protocol changes visibly alter image appearance and reduce segmentation agreement.

Because both datasets consist of healthy participants and include repeat acquisitions with short inter-scan intervals (SRPBS: consecutive-day traveling-subject scans; SIMON: repeated acquisitions of the same subject), we follow common practice in structural MRI test–retest studies and treat true anatomical change over such intervals as negligible compared to acquisition- and pipeline-induced variability. Therefore, in the absence of voxel-wise manual annotations, we interpret agreement metrics as measures of reproducibility/consistency

rather than absolute accuracy (Iscan et al., 2015; Han et al., 2006; Jovicich et al., 2009; Reuter et al., 2012).

### 3.2. Segmentation pipelines

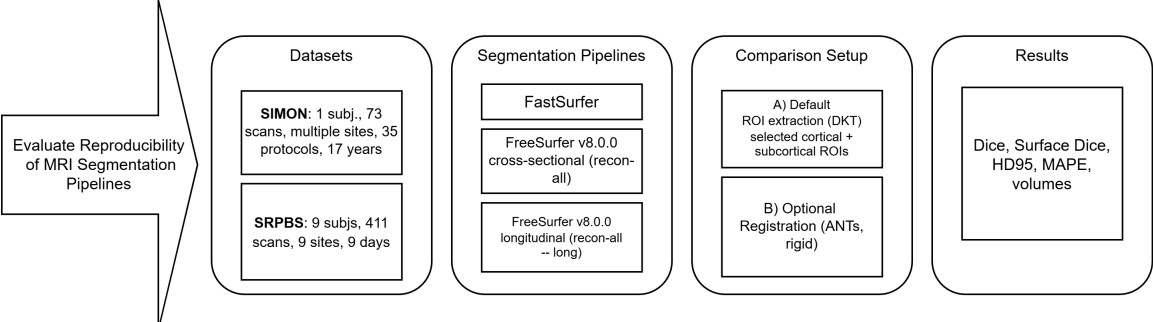

Figure 2: High-level overview of the benchmarking pipeline. The workflow evaluates the reproducibility of FreeSurfer and FastSurfer using the longitudinal SIMON dataset and the multi-site SRPBS traveling-subject dataset. All outputs are mapped to a common ROI space and compared with spatial and volumetric metrics while varying registration and interpolation choices. The benchmark is designed to isolate pipeline-, scanner-, and post-processing-induced variability from true biological change.

We employed FreeSurfer 8.0.0 for cortical surface reconstruction and anatomical segmentation using the `recon-all` pipeline. To evaluate segmentation performance, we compared two state-of-the-art deep learning-based methods: FastSurfer (Henschel et al., 2020b) and SynthSeg (Billot et al., 2023c). FastSurfer offers rapid and accurate whole-brain segmentation, replicating FreeSurfer's anatomical outputs, while SynthSeg provides robust segmentation across varying MRI contrasts and resolutions without the need for retraining. For consistency and comprehensive analysis, we selected FreeSurfer's `recon-all` outputs as the common baseline and assessed the Desikan-Killiany-Tourville (DKT) atlas parcellations, encompassing 100 cortical and subcortical regions.

**Preprocessing and input standardization.** To maximize reproducibility and avoid introducing study-specific choices, we intentionally did not apply additional external preprocessing (e.g., custom intensity normalization) beyond each pipeline's default processing. In prior work on tumor segmentation, the choice of segmentation pipeline was reported to have a negligible effect on segmentation accuracy (Kondrateva et al., 2024). FreeSurfer recon-all includes its canonical internal conforming and intensity processing as part of the standard workflow (Fischl, 2012b). SynthSeg is trained via domain randomization to be robust to wide variations in contrast, resolution, noise and bias fields, and is designed to operate without requiring dedicated preprocessing (Billot et al., 2023c).

For surface-based metrics, we applied rigid-body registration using ANTs (Avants et al., 2011), computing transforms from the original T1-weighted images. We evaluated two in-

terpolation modes: linear and nearest neighbor. Two registration strategies were compared: (1) intra-subject co-registration to the subject's first session for longitudinal alignment, and (2) spatial normalization to an asymmetric MNI atlas. This approach aimed to assess the impact of interpolation schemes and reference space choice on the consistency of surface-derived measurements.

**ROI Analysis.** We focused our analysis on 9 cortical and 8 subcortical bilateral regions of interest (ROIs), selected based on their relevance as biomarkers in neuroimaging studies. The complete list of analyzed ROIs is provided in Table 4. Differences observed across successive MRI sessions were interpreted as domain variations.

### 3.3. Metrics.

To evaluate segmentation reproducibility, we report absolute volume differences, as well as spatial similarity metrics: Dice coefficient, Surface Dice, and 95th percentile Hausdorff Distance (HD95). Each metric captures a different aspect of agreement between two segmentations: volumetric overlap, boundary proximity, and outlier misalignment. These are computed for each region of interest (ROI) and aggregated across sessions. To compare volumes across repeated scans, we use the mean absolute percentage error between segmentation volumes.

**Dice Coefficient (DSC).** DSC (Dice, 1945b) measures the voxel-level overlap between two binary masks $A$ and $B$ (e.g., predicted and reference segmentation):

$$\text{DSC} = \frac{2|A \cap B|}{|A| + |B|} \tag{1}$$

Here, $|A|$ and $|B|$ are the number of voxels in each mask, and $|A \cap B|$ is the number of voxels they share. Dice is widely used due to its simplicity, but can be insensitive to boundary errors.

**Surface Dice (S-DSC).** Surface Dice (Nikolov et al., 2018) quantifies the proportion of surface points that lie within a distance $\tau$ between the two segmentation boundaries $\partial A$ and $\partial B$:

$$\text{S-DSC} = \frac{|\{x \in \partial A : d(x, \partial B) \leq \tau\}| + |\{y \in \partial B : d(y, \partial A) \leq \tau\}|}{|\partial A| + |\partial B|}. \tag{2}$$

Here, $d(x, \partial B)$ denotes the minimum Euclidean distance from a point $x$ on the surface of $A$ to the surface of $B$, and $\tau$ is the distance tolerance (set to $1\,\text{mm}$ in our experiments). This metric captures small surface deviations and is well-suited for assessing perceptual segmentation accuracy.

**95th Percentile Hausdorff Distance (HD95).** HD95 (Huttenlocher et al., 1993) captures the worst-case boundary discrepancy, ignoring extreme outliers by focusing on the 95th percentile of all boundary distances:

$$\text{HD}_{95}(A, B) = \max \left\{ \begin{array}{l} \text{P}_{95}\big(\{d(x, \partial B) : x \in \partial A\}\big), \\ \text{P}_{95}\big(\{d(y, \partial A) : y \in \partial B\}\big) \end{array} \right\}. \tag{3}$$

Where $\text{P}_{95}$ denotes the 95th percentile, and $d(x, \partial B)$ is the shortest distance from point $x$ to the other surface. HD95 is useful for identifying large local deviations in shape or topology.

**Mean Absolute Percentage Error (MAPE):** To compare volumes across repeated scans, we use the mean absolute percentage error between segmentation volumes:

$$\text{MAPE} = \frac{100\%}{n} \sum_{i=1}^{n} \left| \frac{V_i^{\text{pred}} - V_i^{\text{ref}}}{V_i^{\text{ref}}} \right| \tag{4}$$

Where $V_i^{\text{pred}}$ and $V_i^{\text{ref}}$ are the predicted and reference volumes for region $i$, and $n$ is the number of ROIs. MAPE is intuitive for assessing how much segmentations deviate from expected anatomical volumes.

### 3.4. Computation Environment.

We used a single compute environment for all benchmark experiments, reflecting practical deployment constraints and emphasizing reproducibility.

- **CPU morphometry:** FreeSurfer 8.0.0 `recon-all` (cross-sectional and longitudinal) was executed on a Google Cloud Platform (GCP) instance equipped with *64* vCPUs and *512* GB of RAM. FreeSurfer was run using a single CPU core per subject, with an average processing time of approximately $\sim 2$ hours/subject (longitudinal runs typically longer due to within-subject template construction and additional processing steps). Attempts to utilize GPU acceleration for FreeSurfer 8.0.0 were unsuccessful due to driver/library compatibility issues; therefore, all FreeSurfer processing was performed on the CPU.

- **CNN-based segmentation:** FastSurfer and SynthSeg were executed within the same benchmark framework. While these models can benefit substantially from GPU acceleration, we report GPU runtimes as reference from the respective projects: FastSurfer reports whole-brain inference on the order of minutes on GPU and substantially reduced end-to-end runtime compared to classical FreeSurfer pipelines (FastSurfer Developers, 2026).

**Model parameters:** SynthSeg (Billot, 2023) employs a 3D U-Net with five levels, starting from 24 feature maps and doubling at each downsampling layer, corresponding to $\sim 20$–40 million trainable parameters (computed directly from the official Keras/TensorFlow model definition) (Billot et al., 2023c); FastSurferCNN and FastSurferVINN use lightweight U-Net–style fully convolutional networks with only $\sim 1.8$–1.85 million parameters (Henschel et al., 2022); for comparison, browser-based BrainChop leverages a MeshNet-style 3D CNN with dilated convolutions, achieving interactive inference on consumer hardware with a compressed TensorFlow.js model of only tens of megabytes ($\sim 1$–5 million parameters) (Tudosiu et al., 2023).

## 4. Results

We first report short-interval test–retest findings from the SRPBS traveling-subject dataset to isolate scanner- and site-related variability under minimal expected biological change. We then compare longitudinal behavior in the SIMON dataset and conclude with cross-dataset analyses of overlap, volume reproducibility, and the effects of registration and interpolation choices.

### 4.1. SRPBS Test–Retest: FastSurfer

We analyzed 15 sessions from the SRPBS Traveling Subject dataset (Tanaka et al., 2021) using FastSurfer . As shown in Figure 3, the first five sessions were acquired on the same scanner across five consecutive days, while the remaining sessions involved different scanners and sites.

For both hippocampus and amygdala, volume estimates during the same-scanner phase were highly consistent. For example, left hippocampus volumes ranged narrowly between 4.42–4.44 cm$^3$ (SD = 0.01), and right amygdala volumes ranged from 1.73–1.75 cm$^3$ (SD = 0.008). In contrast, sessions from different scanners showed noticeable variability: left hippocampus ranged from 4.16–4.53 cm$^3$ (SD = 0.10), and right amygdala from 1.50–1.85 cm$^3$ (SD = 0.11).

This highlights that even in a highly controlled test-retest design, inter-scanner variability introduces morphometric noise of up to 10%, especially in small structures like the amygdala. Reliable quantification in longitudinal or multisite settings requires either harmonization or robust outlier filtering.

### 4.2. SIMON Longitudinal: FastSurfer vs. FreeSurfer

We evaluated segmentation reproducibility across 73 sessions over 17 years using FastSurfer and FreeSurfer.

**FastSurfer.** FastSurfer `recon-all` failed on 3 sessions and 8 runs. For valid outputs, subcortical volumes were stable: Left/Right Amygdala: $1.93 \pm 0.17$ / $2.10 \pm 0.12$ cm$^3$ Left/Right Hippocampus: $4.54 \pm 0.19$ / $4.82 \pm 0.16$ cm$^3$ Volume trajectories showed small upward trends ($R^2 = 0.12$–0.26).

**FreeSurfer.** Subcortical variation averaged 3.1%, peaking at 15–20%. Cortical parcellations varied by 5% on average, with outliers exceeding 40–90%. Volumes were consistently higher: Amygdala: $2.13 \pm 0.07$ / $2.22 \pm 0.07$ cm$^3$ Hippocampus: $5.10 \pm 0.11$ / $5.18 \pm 0.12$ cm$^3$.

Volume comparisons show that FreeSurfer consistently estimates larger volumes than FastSurfer. For example, the left hippocampus volume averaged $5.10 \pm 0.11$ cm$^3$ in FreeSurfer versus $4.54 \pm 0.19$ cm$^3$ in FastSurfer.

Table 1 compares FreeSurfer and FastSurfer across eight representative cortical structures. FastSurfer yielded consistently higher Dice scores (e.g., 0.861 vs. 0.793 for Insula, 0.816 vs. 0.728 for Fusiform), suggesting improved anatomical overlap. Surface Dice values remained comparable, with minimal variation between methods. Volume differences were notably smaller in FastSurfer (e.g., 2.0 mm$^3$ for Insula, compared to 31.6 mm$^3$ in FreeSurfer), reflecting reduced bias. Interestingly, FreeSurfer produced lower Hausdorff distances in some regions (e.g., Superior Frontal Cortex: 1.21 mm vs. 1.74 mm), but at the cost of greater volume deviation. Overall, FastSurfer offers more consistent cortical segmentation while maintaining competitive boundary accuracy.

### 4.3. Comparison of Distance-based Metrics Across Datasets

Volume differences (in cm$^3$) were consistently higher in SRPBS. In contrast, SIMON—being a single-subject longitudinal dataset—showed lower volume deviations across repeated scans. Dice and Surface Dice scores were uniformly higher in SIMON, indicating improved

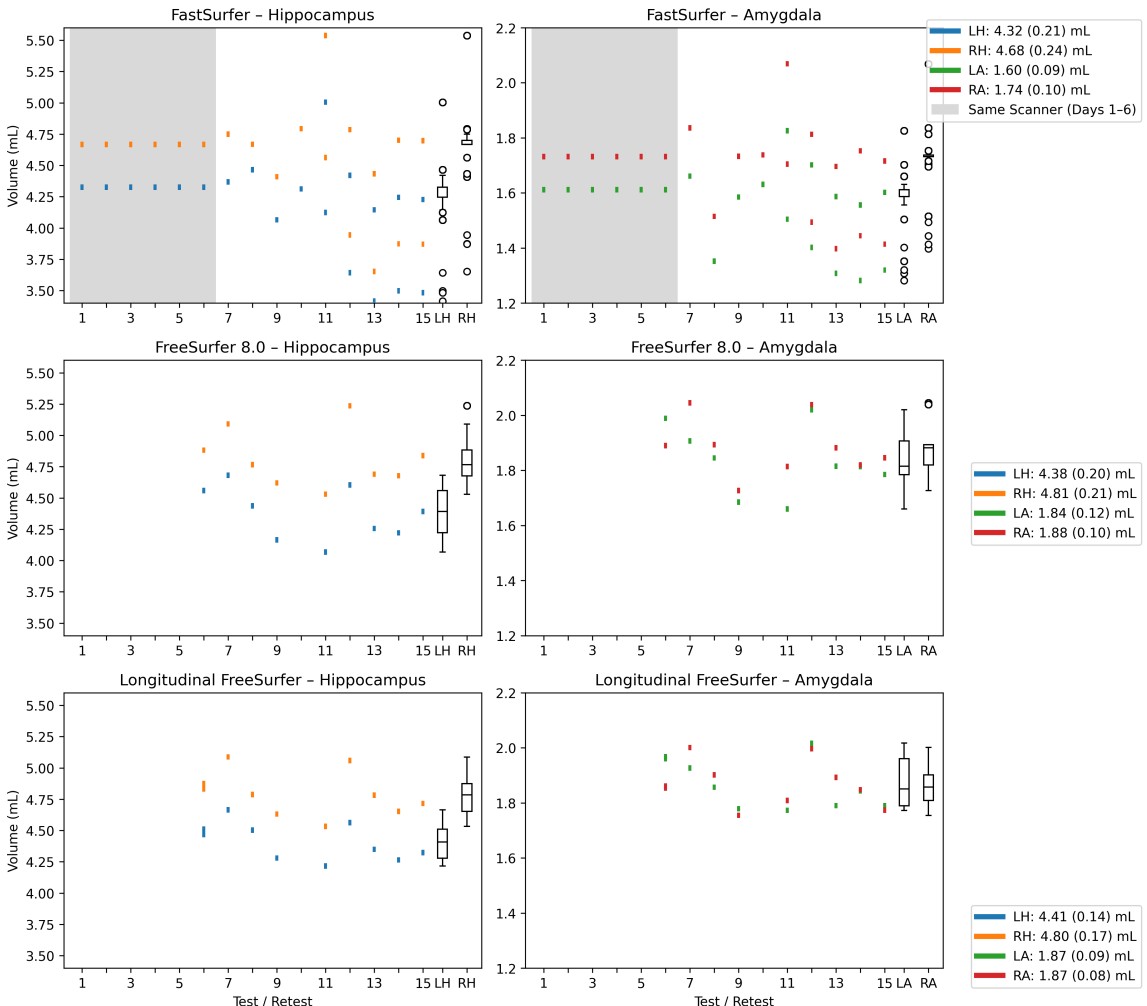

Figure 3: Volume estimates for left and right hippocampus and amygdala across 15 scans of the same traveling subject in the SRPBS dataset (sub-01), processed with Fast-Surfer, FreeSurfer 8, and longitudinal FreeSurfer 8 with creation of the unbiased template. Each point corresponds to one session; the shaded region marks the first five scans acquired on a single scanner (days 1–6), whereas the remaining sessions were collected at different sites. Within-scanner repeat scans are tightly clustered, but cross-site acquisitions introduce much larger spread, especially in the amygdala, showing that scanner effects can approach the size of subtle longitudinal change.

overlap and surface-level agreement. For example, mean Dice scores for the caudate and putamen reached 0.868 and 0.897 in SIMON, compared to 0.802 and 0.848 in SRPBS. HD95 distances also decreased in SIMON (e.g., 1.234 mm for hippocampus vs. 1.830 mm

Table 1: Comparison of FreeSurfer 8 (FS) and FastSurfer (Fast) segmentation performance across subcortical structures. Volume differences are in mm$^3$, Dice and Surface Dice are unitless, HD95 is in mm.

| Metric | Accumbens | | Amygdala | | Caudate | | Hippocampus | | Pallidum | | Putamen | | Thalamus | | Ventral DC | |
|---|---|---|---|---|---|---|---|---|---|---|---|---|---|---|---|---|
| | FS | Fast | FS | Fast | FS | Fast | FS | Fast | FS | Fast | FS | Fast | FS | Fast | FS | Fast |
| Volume Diff (mm$^3$) | 5.20 | -0.56 | 0.22 | -2.23 | 14.18 | 1.36 | 12.46 | -0.17 | 11.99 | 1.94 | 19.99 | -5.04 | 2.27 | 8.30 | 12.32 | 1.06 |
| Dice | 0.803 | 0.827 | 0.858 | 0.862 | 0.868 | 0.874 | 0.850 | 0.868 | 0.850 | 0.859 | 0.897 | 0.902 | 0.909 | 0.917 | 0.858 | 0.873 |
| Surface Dice | 0.965 | 0.955 | 0.961 | 0.944 | 0.972 | 0.957 | 0.964 | 0.963 | 0.958 | 0.927 | 0.969 | 0.956 | 0.947 | 0.948 | 0.959 | 0.950 |
| HD95 (mm) | 1.23 | 1.60 | 1.26 | 1.50 | 1.20 | 1.56 | 1.23 | 1.34 | 1.27 | 1.64 | 1.21 | 1.58 | 1.33 | 1.45 | 1.23 | 1.43 |

Table 2: Comparison of FreeSurfer 8 (FS) and FastSurfer segmentation performance across selected cortical structures. Volume difference is in mm$^3$, Dice and Surface Dice are unitless, HD95 is in mm.

| Metric | Caudal Ant. Cingulate | | Entorhinal Cortex | | Fusiform Gyrus | | Inferior Parietal | | Insula | | Lat. Orbitofrontal | | Med. Orbitofrontal | | Superior Frontal | | Superior Temporal | |
|---|---|---|---|---|---|---|---|---|---|---|---|---|---|---|---|---|---|---|
| | FS | Fast | FS | Fast | FS | Fast | FS | Fast | FS | Fast | FS | Fast | FS | Fast | FS | Fast | FS | Fast |
| Volume Diff | 21.62 | 2.90 | 7.75 | -4.78 | 58.67 | -2.95 | 113.35 | 3.51 | 31.60 | 2.00 | 64.48 | 7.46 | 35.29 | 5.04 | 216.32 | 42.99 | 115.34 | 15.80 |
| Dice | 0.746 | 0.820 | 0.709 | 0.728 | 0.728 | 0.816 | 0.726 | 0.807 | 0.793 | 0.861 | 0.712 | 0.796 | 0.663 | 0.780 | 0.733 | 0.807 | 0.759 | 0.817 |
| Surface Dice | 0.965 | 0.958 | 0.922 | 0.922 | 0.959 | 0.964 | 0.973 | 0.963 | 0.970 | 0.971 | 0.952 | 0.948 | 0.938 | 0.949 | 0.970 | 0.966 | 0.964 | 0.958 |
| HD95 | 1.24 | 1.64 | 1.72 | 1.72 | 1.28 | 1.35 | 1.19 | 1.47 | 1.35 | 1.51 | 1.34 | 1.85 | 1.46 | 1.84 | 1.21 | 1.74 | 1.26 | 1.47 |

in SRPBS). These results support the utility of repeated intra-subject data for evaluating segmentation consistency.

Statistical comparison using Mann-Whitney U tests with Benjamini-Hochberg FDR correction confirmed significant differences between SIMON and SRPBS conditions for 89% of ROI-metric combinations (64/72, $p_{\text{adj}} < 0.05$). Effect sizes were predominantly large (Cliff's $\delta > 0.474$ in 56/72 comparisons), with the most pronounced differences observed for Dice and Surface Dice in cortical regions (e.g., inferior parietal: $\delta = 0.81$–$0.84$; fusiform gyrus: $\delta = 0.74$–$0.80$). These results provide quantitative evidence that cross-scanner variability introduces systematic and substantial degradation in segmentation consistency.

Contrary to expectations based on study design and field strength, the observed segmentation variability did not align with either nominal time interval or magnet strength: although SIMON (a longitudinal 1.5T dataset) could plausibly exhibit larger true volumetric changes over time and might be expected to be less robust than SRPBS (a consecutive-day 3T traveling-subject dataset), we instead observed generally higher consistency in SIMON and lower consistency in SRPBS. This suggests that acquisition-related domain shift, driven by scanner- and protocol-specific factors and their interaction with each pipeline, dominates

| Metric | Accumbens | | Amygdala | | Caudate | | Hippocampus | | Pallidum | | Putamen | | Thalamus | | Ventral DC | |
|---|---|---|---|---|---|---|---|---|---|---|---|---|---|---|---|---|
| | SRPBS | SIMON | SRPBS | SIMON | SRPBS | SIMON | SRPBS | SIMON | SRPBS | SIMON | SRPBS | SIMON | SRPBS | SIMON | SRPBS | SIMON |
| Volume diff (cm$^3$) | 0.046 | 0.030 | 0.102 | 0.076 | 0.119 | 0.098 | 0.207 | 0.125 | 0.102 | 0.095 | 0.206 | 0.136 | 0.450 | 0.374 | 0.219 | 0.141 |
| Dice | 0.677 | 0.803 | 0.790 | 0.858 | 0.802 | 0.868 | 0.782 | 0.850 | 0.789 | 0.850 | 0.848 | 0.897 | 0.868 | 0.909 | 0.806 | 0.858 |
| Surface Dice | 0.849 | 0.965 | 0.840 | 0.961 | 0.868 | 0.972 | 0.845 | 0.964 | 0.843 | 0.958 | 0.870 | 0.969 | 0.820 | 0.947 | 0.873 | 0.959 |
| HD95 (mm) | 1.735 | 1.228 | 1.697 | 1.263 | 1.584 | 1.200 | 1.830 | 1.234 | 1.675 | 1.271 | 1.582 | 1.210 | 1.828 | 1.327 | 1.620 | 1.233 |

Table 3: Comparison of segmentation metrics between the SRPBS and SIMON datasets across subcortical structures. Volume difference is shown in cm$^3$, Dice and Surface Dice are unitless similarity scores, and HD95 represents the 95th percentile Hausdorff distance in millimeters.

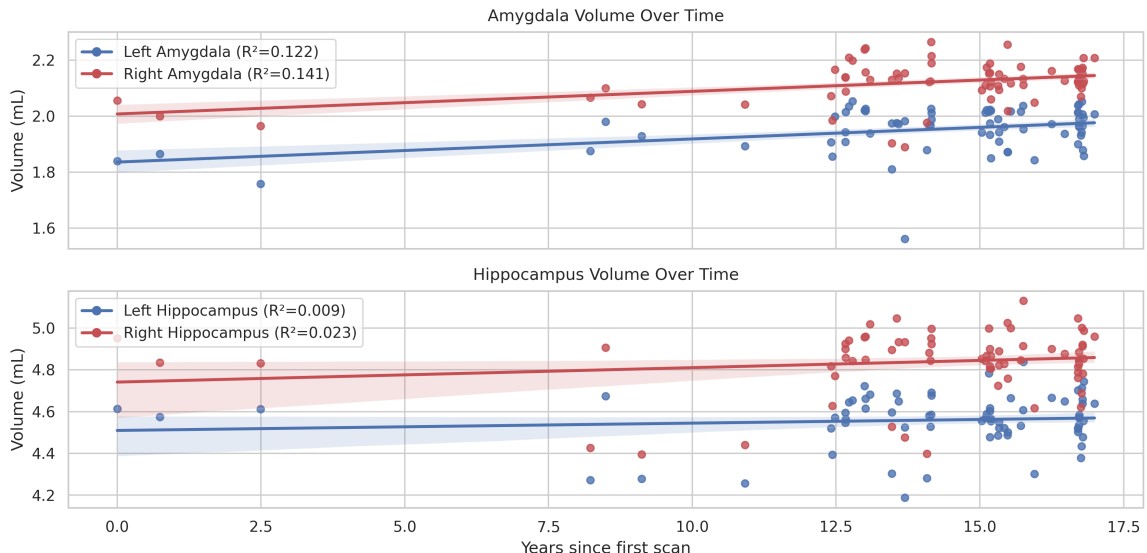

Figure 4: SIMON dataset: volume trajectories of the amygdala and hippocampus over time for 73 MRI scans acquired over 17 years in one healthy individual using FastSurfer. Confidence intervals and regression trends are shown. FastSurfer yields comparatively smooth long-term trajectories with modest drift, suggesting that most within-subject variation is small relative to the overall volume scale.

the variability in these datasets, outweighing both short-term biological change and the presumed robustness advantage of 3T over 1.5T.

**HD95 Stability Analysis.** Hausdorff distance at the 95th percentile (HD95) revealed striking differences in boundary consistency between datasets. For SIMON, HD95 values were remarkably stable: the median was 1.0 mm across most cortical structures, with IQR = 0 for 16/18 regions. Values of exactly 1.0, $\sqrt{2} \approx 1.41$, and $\sqrt{3} \approx 1.73$ mm correspond to 1-, 2D-diagonal, and 3D-diagonal voxel distances at 1 mm isotropic resolution, indicating that segmentation boundaries differ by at most 1–2 voxels between consecutive sessions.

In contrast, SRPBS exhibited substantially higher HD95 variability (median 1.4–2.2 mm, IQR up to 1.9 mm). Notably, certain site combinations produced outlier values reaching 7–9 mm (e.g., siteHKH vs. siteHUH), while others remained at 1.0 mm. This site-dependent variability suggests that specific scanner or protocol combinations introduce systematic boundary inconsistencies, even when comparing the same subject.

**Subcortical Filtering Based on Segmentation Quality.** To assess the impact of quality-based filtering, we evaluated the proportion of subcortical structures removed using various thresholds on Dice and Surface Dice metrics. Applying a strict Surface Dice threshold of 0.92 filtered out only 5% of regions, while retaining a low mean absolute percentage error (MAPE) across the remaining structures (2.8% at 75th percentile, 8.6% at 95th). Relaxing the threshold to 0.90 slightly reduced filtering (3.8%) without degrading MAPE. In contrast, filtering with a traditional Dice threshold of 0.80 excluded more than half of

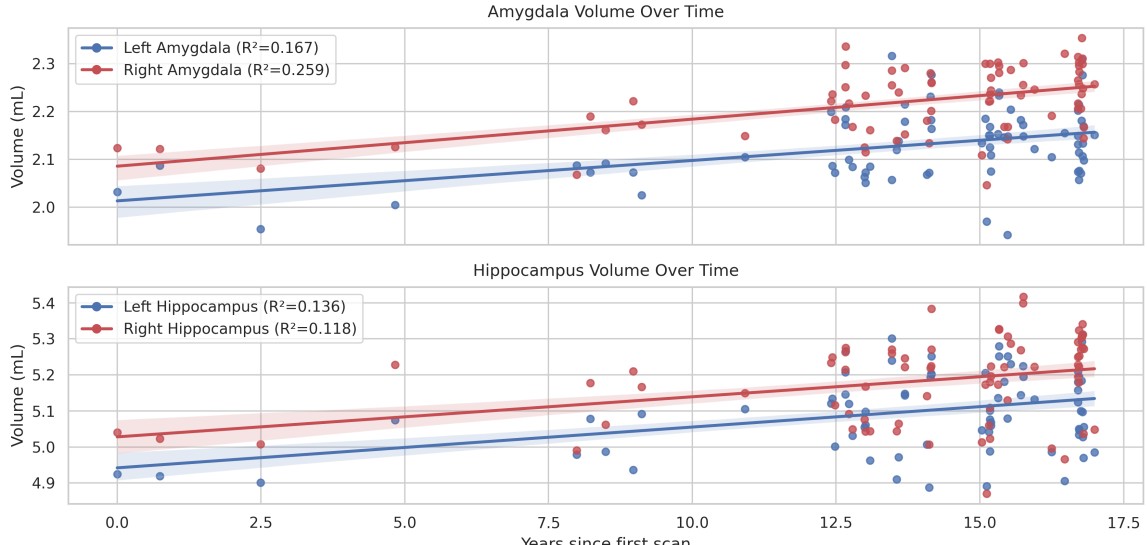

Figure 5: SIMON dataset: volume trajectories of the amygdala and hippocampus over time for 73 MRI scans acquired over 17 years in one healthy individual using FreeSurfer 8. Confidence intervals and regression trends are shown. The inferred longitudinal trend depends on the segmentation software, not only the anatomy: FreeSurfer 8 produces systematically larger estimates and visually broader dispersion than the corresponding FastSurfer trajectories.

all structures (52.8%), yet retained comparable or worse error profiles. This supports the use of Surface Dice as a more efficient and precise filtering criterion for detecting outliers in automated segmentation pipelines.

Table 4: Percentage of subcortical regions filtered out using Dice and Surface Dice thresholds, with 75th and 95th percentile MAPE values across retained regions.

| Filtering Metric | Threshold | Structures | % Filtered | 75th (% MAPE) | 95th (%) |
|---|---|---|---|---|---|
| Surface Dice | 0.92 | Subcortical | 5.0 | 2.8 | 8.6 |
| Surface Dice | 0.90 | Subcortical | 3.8 | 2.8 | 8.8 |
| Dice | 0.80 | Subcortical | 52.8 | 2.2 | 5.8 |

## 4.4. Volume Reproducibility (ICC Analysis)

To quantify the reproducibility of volumetric estimates, we computed intraclass correlation coefficients (ICC(3,1), two-way mixed effects, single measurement, consistency) for cortical ROIs across repeated measurements. For the SIMON dataset, ICC values were predominantly poor (mean ICC = 0.14, range: −0.04 to 0.32), reflecting high inter-scanner variability in volume estimates despite reasonable segmentation overlap. This finding high-

lights that even when segmentation boundaries are consistent (Dice ∼0.80), absolute volume estimates can vary substantially across different scanners and protocols in time.

In contrast, the SRPBS dataset exhibited moderate ICC values (mean ICC = 0.68, range: 0.42–0.87), with 8/18 structures achieving good reliability (ICC > 0.75). The higher ICC in SRPBS, despite lower Dice scores, suggests that the traveling-subject protocol with controlled acquisition parameters produces more consistent volume estimates across sites than the heterogeneous SIMON acquisitions spanning 17 years and 35 different protocols.

These findings underscore a critical distinction: segmentation *overlap* (Dice) and volume *reproducibility* (ICC) capture different aspects of reliability. Cross-scanner variability primarily affects absolute volume quantification rather than anatomical boundary delineation.

### 4.5. Registration Strategies and Interpolation Effects

To compare surface-based metrics, rigid-body registration was applied using ANTs (Avants et al., 2011). We tested two interpolation strategies, `nearestNeighbor` and `genericLabel`, and two reference spaces: subject-native (first session) and standard MNI atlas. Interpolation mode affected mean volume estimates by up to 1.72%, while template choice accounted for a smaller 0.07% deviation.

## 5. Conclusion and Discussion

This study shows that even state-of-the-art segmentation tools such as FastSurfer and FreeSurfer remain sensitive to scanner and protocol variability, particularly in multi-site and longitudinal settings. Despite their wide adoption and strong benchmark performance, we observed non-negligible instability in repeated measurements, especially for small subcortical structures such as the amygdala and pallidum.

On the SRPBS Traveling Subject dataset, both tools achieved excellent within-scanner consistency over five consecutive days, with volume deviations below 1%. However, cross-site sessions for the same individual and nominal protocol produced fluctuations up to 10%, directly constraining the ability to detect subtle longitudinal changes in small structures. In the 17-year longitudinal SIMON dataset, both FastSurfer and FreeSurfer exhibited increasing volume trends over time, but differed in the magnitude and smoothness of these trajectories, indicating method-dependent biases in long-term morphometric estimates.

FreeSurfer systematically yielded larger subcortical volumes than FastSurfer (e.g., left hippocampus: $5.10\,\mathrm{cm}^3$ vs. $4.58\,\mathrm{cm}^3$) and showed greater inter-scan variation in cortical regions. Such systematic offsets and differential variability imply that absolute volumes and longitudinal slopes are not directly interchangeable between methods. In practical terms, this reinforces the need for harmonization strategies, method-specific calibration, or stringent quality-control filters in real-world neuroimaging pipelines.

We quantified reliability using a set of complementary overlap and distance-based metrics rather than relying on a single indicator. This multi-metric approach captures distinct aspects of segmentation behaviour, such as boundary stability, volumetric agreement, and sensitivity to outlier scans, and provides a more informative picture of robustness than Dice alone.

While recent work in brain MRI segmentation has focused on speed, automation, and ease of deployment, our results suggest that robustness to scanner and protocol variation

remains a primary bottleneck for individualized applications. We release a lightweight, fully reproducible evaluation pipeline on longitudinal and multi-scanner datasets, with the aim of encouraging more transparent and method-agnostic benchmarking of segmentation tools under conditions that approximate real clinical and research use, in line with recent decentralized benchmarking efforts in healthcare AI such as the FeTS challenge (Zenk et al., 2025).

## 5.1. Limitations

**Reference Annotations.** Both SRPBS and SIMON datasets lack manual annotations, preventing true accuracy assessment (Kofler et al., 2023). We evaluated reproducibility under the assumption that anatomical structures remain stable in healthy subjects. We use FreeSurfer `recon-all` as a *reference label space* (not as ground truth) to enable like-for-like ROI definitions and surface-based morphometry across pipelines. In the absence of manual voxel-wise annotations, our evaluation targets *reproducibility/consistency* rather than absolute accuracy, and conclusions are based on relative variability patterns across ROIs and sites.

A further limitation is that the available datasets are not ideal for isolating scanner effects: SRPBS provides a traveling-subject design but includes only nine subjects, and SIMON is a single-subject longitudinal dataset. These small sample sizes limit statistical power and may reduce the generalizability of conclusions about inter-scanner and longitudinal variability patterns.

**Preprocessing and Augmentation.** We processed raw data without denoising, intensity normalization, or augmentation to isolate the effect of domain shift. Although this reflects practical variability, it limits reproducibility. It has been shown that classical preprocessing techniques, such as intensity normalization and histogram matching, do not consistently improve brain tumor segmentation performance across different domains. This limitation underscores the challenges posed by domain shifts in medical imaging. However, recent advancements in generative methods, including those utilizing generative adversarial networks (GANs), offer promising avenues to address these challenges. For instance, methods like M-GenSeg employ semi-supervised generative training strategies for cross-modality tumor segmentation, demonstrating improved generalization across diverse imaging modalities (Alefsen de Boisredon d'Assier et al., 2022). Related brain-MRI domain-adaptation work has also explored Fader networks on ABIDE-II fMRI, reinforcing that representation-level adaptation may help when acquisition domains differ (Pominova et al., 2021).

**Software and model choice.** We did not compare against an established FreeSurfer baseline such as v7.4 (the benchmark uses FreeSurfer 8.0.0), and we did not evaluate Fast-Surfer in longitudinal mode (`--long`) alongside its cross-sectional pipeline; additionally, although both T1 and T2 were available for many (but not all) sessions, we did not assess multi-contrast segmentation/parcellation or contrast-synthesis to enable consistent multimodal inputs across all sessions, even though modality-ablation studies in multimodal brain MRI segmentation suggest that the contribution of each sequence can materially affect baseline performance and transferability (Druzhinina et al., 2022); nor did we include additional state-of-the-art segmentation families (e.g., nnU-Net/nnFormer) or foundation/VLM-style

approaches (e.g., MedSAM), which may limit generalization of our conclusions beyond the selected pipelines and input setting.

## 5.2. Future Directions

1. Software, model, and configuration combinations for robustness. Systematically evaluate robustness-oriented "pipelines of pipelines" rather than single tools: FreeSurfer or Fast-Surfer and more combined with alternative skull-stripping, intensity standardization, and surface/cortical post-processing choices, and quantify which combinations reduce variability without sacrificing anatomical plausibility. In particular, test inference-time strategies such as test-time augmentation, uncertainty-based filtering, and consensus or ensemble labeling to stabilize ROI estimates across scanners and sessions.

2. Extend the benchmark to include low-field scanner acquisitions, where SNR, bias fields, motion, and resolution differ substantially from standard research-grade protocols. This would enable stress-testing segmentation robustness under realistic clinical constraints and help identify which methods and QC thresholds remain reliable when acquisition quality is degraded or highly variable.

3. While we focus on T1-weighted brain MRI segmentation, the proposed benchmark design is broadly applicable to other anatomical segmentation settings (e.g., different organs, other MRI contrasts, or CT), whenever repeated-measurement data are available (traveling-subject, scan–rescan, or longitudinal acquisitions). Beyond segmentation, the same repeated-measurement framework can be used as a practical QC instrument to monitor scanner drift and protocol changes over time, and to quantify robustness of downstream modules such as registration, interpolation, and ROI extraction under domain shift. Finally, our evaluation protocol can serve as a consistent testbed for comparing harmonization strategies (e.g., feature-level harmonization such as ComBat (Fortin et al., 2018), image-level normalization, or generative image translation) using the same reproducibility metrics. Adapting the benchmark to a new application primarily requires replacing the ROI/label definitions (atlas) and, where appropriate, adjusting metric tolerances (e.g., surface-distance thresholds) to reflect region size and target anatomy; the core idea of quantifying variability under repeated measurements remains unchanged.

## CRediT authorship contribution statement

**Ekaterina Kondrateva**: Conceptualization, Methodology, Software, Formal analysis, Investigation, Writing – original draft, Writing – review and editing.
**Sandzhi Barg**: Software, Data curation, Validation, Visualization, Writing – original draft, Writing – review and editing.
**Florian Kofler**: Methodology, Validation, Writing – review and editing (reviewed methodological design and manuscript drafts).
**Abdalla Z. Mohamed**: Methodology, Validation, Writing – review and editing (reviewed methodological design and manuscript drafts; provided critical assessment of missing methodological components, including the longitudinal FreeSurfer analysis).

## Declaration of Generative AI and AI-assisted technologies in the writing process

During the preparation of this work, the authors used ChatGPT to improve the readability of this paper. After using this tool, the authors reviewed and edited the content as needed and take full responsibility for the content of the publication.

## Acknowledgments

This work was supported by Anna Valentina Lioba Eleonora Claire Javid Mamasani and the Gemeinnützige Hertie Stiftung. Dr. Abdalla Z Mohamed is supported by CHSS internal fund from United Arab Emirates University (Grant ID:12H115).

We thank Martin Reuter (DZNE) and Thomas Kirk (Quantified Imaging) for their invaluable comments on the preprint version.

We thank Mikhail Vasiliev for help with data visualization (see Appendix A).

This material is based upon work supported by the Google Cloud Research Credits program under award GCP19980904.

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

# Appendix A. Related Work

## A.1. Software Usage for Test–Retest Brain MRI Morphometry and Segmentation

Test–retest designs are commonly used to quantify the *measurement reliability* of structural brain MRI morphometry and segmentation pipelines, isolating biological effects from variability induced by acquisition, preprocessing, and algorithmic choices. Importantly, reliability (repeatability across repeated scans) is distinct from accuracy (agreement with a reference delineation): stable measurements may still be inaccurate, especially in the presence of pathology or large lesions (Eggert et al., 2012). Reliability is typically reported using intraclass correlation coefficients (ICC) (Shrout and Fleiss, 1979) (and related consistency metrics), whereas overlap-based agreement with a reference labeling is often summarized using the Dice coefficient (Dice, 1945a).

A substantial share of the research literature relies on the `FreeSurfer recon-all` pipeline as a *de-facto baseline* for surface-based morphometry and whole-brain parcellation, largely due to its early introduction, broad adoption, and comparability with prior work. The core surface-based framework was established through cortical surface reconstruction and coordinate-system formulations (Dale et al., 1999; Fischl et al., 1999; Sereno et al., 1996) and consolidated in the widely used `FreeSurfer` software description (Fischl, 2012a), including cortical thickness estimation (Fischl and Dale, 2000). Test–retest studies have examined `FreeSurfer` reliability within-site and between-site, showing that visual QC and approval criteria can materially affect reproducibility estimates (Iscan et al., 2015). More recent work also evaluated reliability across acquisition sequences (e.g., MPRAGE vs. MP2RAGE), highlighting protocol-dependent effects on derived volumes, areas, and thickness (Tran et al., 2022).

Beyond surface-based processing, voxel-based morphometry (VBM) remains a widely used alternative, supported by `SPM12` (Wellcome Centre for Human Neuroimaging, 2014) and `CAT` (Gaser and Dahnke, 2016; Gaser et al., 2024), with classic large-scale VBM studies demonstrating sensitivity to aging effects (Good et al., 2001). The literature also includes multi-atlas segmentation approaches, grounded in extensive methodological work (Iglesias and Sabuncu, 2015) and implemented in toolchains such as `MUSE` (Doshi et al., 2016). In multi-site aging contexts, comparative analyses have reported differences between `FreeSurfer` and multi-atlas methods in terms of size/age bias and inter-scanner stability (Srinivasan et al., 2020). More recently, deep learning pipelines have been adopted to improve throughput and robustness: `FastSurfer` provides a high-throughput deep learning alternative (Henschel et al., 2020a), with subsequent work addressing resolution-independence (Henschel et al., 2022) and specialized sub-segmentation tasks (Estrada et al., 2023). In parallel, `SynthSeg` was proposed to enable segmentation across heterogeneous clinical MRI contrasts and resolutions without retraining (Billot et al., 2021, 2023b), and later scaled for large heterogeneous datasets (Billot et al., 2023a). Tool and atlas choices can substantially affect downstream morphometric conclusions, motivating explicit reporting of atlas/tool decisions (Hammers et al., 2020) and broader syntheses of publicly available segmentation methods (Pham et al., 2021).

Recent `FreeSurfer` releases have also modernized the ecosystem by distributing and integrating deep learning modules (e.g., `SynthStrip`, `SynthSeg`, `SynthMorph`) within the

broader `recon-all` workflow (FreeSurfer Development Team, 2024). Given that `SynthSeg` targets robustness to domain shift and acquisition heterogeneity (Billot et al., 2023b,a), these updates are expected to further increase uptake in research settings where multi-site and real-world clinical variability is a primary concern.

Finally, tool usage diverges in pathology-specific contexts. Brain tumor segmentation typically relies on dedicated lesion segmentation methods and challenge-driven toolkits (e.g., BraTS and clinical translation efforts) rather than general morphometry pipelines (Lavoie et al., 2020; BraTS Challenge Organizers, 2024a,b; Hasan et al., 2023). In temporal lobe epilepsy (TLE), hippocampal/MTL segmentation is commonly validated in clinical cohorts and may leverage specialized imaging contrasts and protocols beyond standard T1-based morphometry (Winston et al., 2018; Peixoto et al., 2024). In dementia-oriented volumetry, narrative reviews continue to report substantial reliance on `FreeSurfer`, reflecting its enduring role as a research baseline (Khadhraoui et al., 2024), while newer deep learning and atlas resources expand the methodological landscape (Casamitjana et al., 2025).

## A.2. Usability in Research Practice: Operational Factors Shaping Tool Choice

In research practice, the "usability" of morphometry and segmentation software is determined not only by nominal accuracy or reported reliability, but also by operational burden and reproducibility at scale. Typical workflows are batch pipelines (preprocessing $\rightarrow$ segmentation/surface reconstruction $\rightarrow$ feature extraction $\rightarrow$ statistical analysis), where usability depends on version stability, automation of consistent parameterization, and robustness to heterogeneous data. Related work outside morphometry-specific benchmarking has likewise emphasized the importance of reproducible validation pipelines (Kondrateva et al., 2020) and practical software ecosystems for MRI data analysis (Bernstein et al., 2018). For surface-based pipelines, visual QC is often essential and can meaningfully affect test–retest outcomes, motivating explicit reporting of QC/approval procedures (Iscan et al., 2015).

Compute cost and runtime are additional practical constraints. Computationally intensive classical pipelines have motivated accelerated alternatives and deep learning-based replacements in high-throughput studies (Henschel et al., 2020a). Likewise, robustness to heterogeneous clinical data (variable contrast, resolution, and quality) is increasingly treated as a usability requirement; methods explicitly designed for such heterogeneity (e.g., `SynthSeg`) are therefore increasingly used as pragmatic components in research pipelines (Billot et al., 2023b,a). These operational factors—QC burden, heterogeneity sensitivity, compute cost, and version stability—are especially consequential in test–retest and multi-site settings.

## A.3. Practical Guidelines for Researchers

Based on our limited analysis of segmentation stability, we offer the following guidelines for researchers using automated brain parcellation as per Jan 2026:

**Scanner manufacturer effects.** Cross-manufacturer comparisons (e.g., Siemens vs. GE) produce substantially degraded metrics. In our SRPBS dataset, comparing scans from HKH (Siemens Spectra) and HUH (GE Signa HDxt) yielded HD95 values 7–8$\times$ higher than same-manufacturer comparisons (e.g., 7.81 mm vs. 1.0 mm for caudal anterior cingulate). For

multi-site studies, we recommend stratifying analyses by scanner manufacturer or applying harmonization methods before pooling data.

**Software selection.** FastSurfer demonstrated superior volume consistency (bias reduced 5–10×) and higher Dice scores for cortical structures (+8–12% improvement over FreeSurfer). FreeSurfer showed marginally better boundary precision (HD95 approximately 0.3 mm lower). For volumetric studies, FastSurfer is preferred; for cortical thickness or atrophy studies where boundary precision is critical, FreeSurfer remains a valid choice despite longer processing time.

**Structure-specific reliability.** Cortical structures showed substantial variability in reproducibility. The most stable regions were superior temporal (Dice = 0.849) and superior frontal cortex (Dice = 0.823), while the least stable were lateral orbitofrontal (Dice = 0.734) and entorhinal cortex (Dice = 0.757). We recommend applying stricter QC thresholds (Surface Dice > 0.90) for entorhinal and orbitofrontal regions, and validating findings involving these structures with manual review.

**QC Recommendations.** Our analysis reveals that universal Surface Dice thresholds may be inappropriate for quality control, as structure-specific variability ranges substantially—from 5th percentile values of 0.50 (caudal anterior cingulate) to 0.93 (ventral diencephalon). We recommend structure-specific thresholds based on the 10th percentile of the reference distribution: segmentations falling below this threshold should be flagged for manual review. The complete table of empirically derived failure and acceptability thresholds for all analyzed structures is available in our code repository.

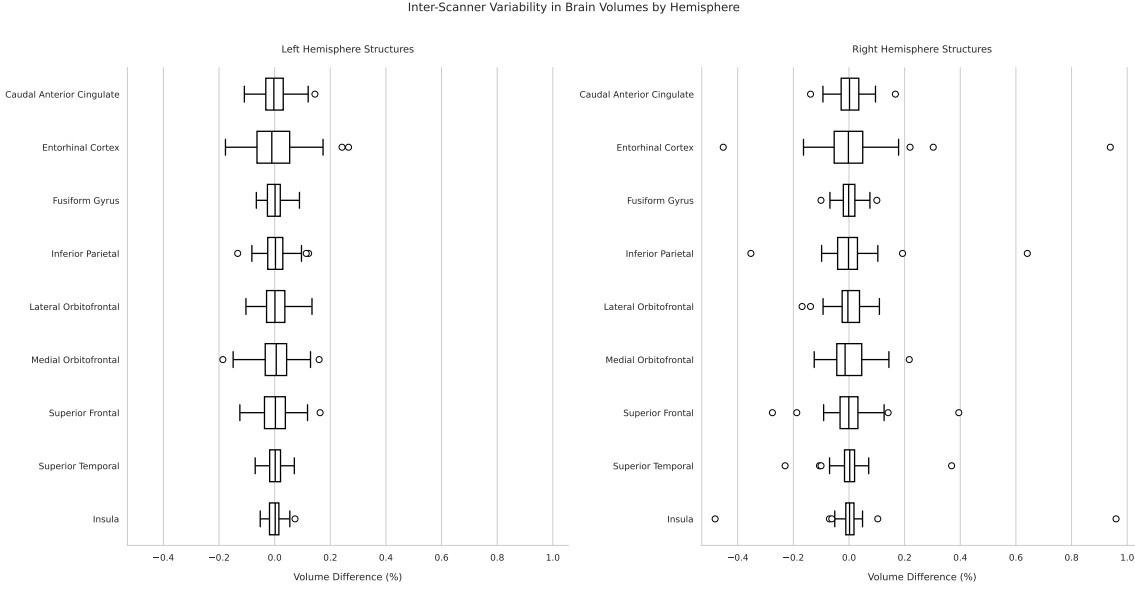

Figure 6: Inter-scanner variability of cortical volumes in the SIMON dataset. Boxplots show the percentage difference from the structure-specific mean across repeated sessions, grouped by hemisphere. Cortical volume reproducibility is strongly region-dependent, with some regions showing much wider spread and outliers than others, so a single global QC threshold is unlikely to be sufficient.

| Tool / Family | Year | Key reference | Typical research usage (test–retest / morphometry) |
|---|---|---|---|
| FreeSurfer (`recon-all`) | 1999–2012; 2025 update | Dale et al. (1999); Fischl et al. (1999); Fischl (2012a); FreeSurfer Development Team (2024) | De-facto baseline for surface-based morphometry and parcellation; test–retest reliability depends on QC/approval (Iscan et al., 2015; Tran et al., 2022). **v8 adds Synth\* DL modules (incl. SynthSeg)** (FreeSurfer Development Team, 2024), likely increasing adoption for domain-robust processing (Billot et al., 2023a). |
| SPM12 (VBM) | 2014 | Wellcome Centre for Human Neuroimaging (2014) | VBM workflows for group-level morphometry; common alternative to surface-based processing (Good et al., 2001). |
| CAT (VBM toolbox) | 2016–2024 | Gaser and Dahnke (2016); Gaser et al. (2024) | VBM-oriented toolbox; widely used for streamlined preprocessing and morphometric inference. |
| FSL (structural suite) | 2000 | FMRIB Analysis Group (2000) | Structural preprocessing/segmentation components used in broader workflows; sometimes included in reproducibility comparisons. |
| MUSE (multi-atlas) | 2016 | Doshi et al. (2016) | Multi-atlas ROI segmentation; compared to FreeSurfer in multi-site aging with reports on stability and bias (Srinivasan et al., 2020). |
| FastSurfer (DL pipeline) | 2020–2021 | Henschel et al. (2020a, 2022) | High-throughput DL alternative producing FreeSurfer-like outputs; used when runtime and scalability dominate. |
| SynthSeg (robust DL) | 2021–2023 | Billot et al. (2021, 2023b,a) | Designed for heterogeneous clinical MRI without retraining; adopted for domain robustness and large-scale analyses. |
| NeuroQuant (clinical volumetry) | 2007 | Cortechs.ai (2007) | Clinical-facing volumetry; sometimes used in research for clinical comparability. |
| BraTS ecosystem (tumor) | 2020–2024 | Lavoie et al. (2020); BraTS Challenge Organizers (2024a,b) | Dedicated lesion segmentation workflows for neuro-oncology; distinct from general morphometry pipelines (Hasan et al., 2023). |
| TLE hippocampal / MTL | 2018–2024 | Winston et al. (2018); Peixoto et al. (2024) | Epilepsy-focused hippocampal/MTL morphometry with clinical validation and specialized protocols/contrasts. |
| NextBrain (atlas) | 2025 | Casamitjana et al. (2025) | Histology-informed probabilistic atlas; emerging use for fine-grained MRI analysis. |

Table 5: Publication timeline and typical research usage patterns for common brain MRI morphometry and segmentation toolchains in test–retest and related study designs.

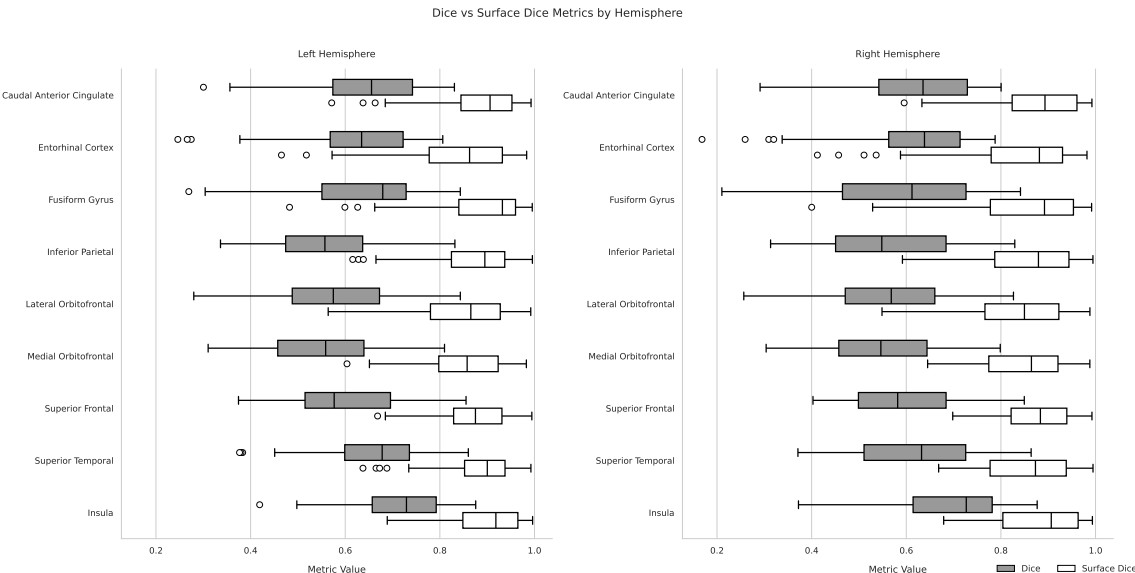

Figure 7: Inter-scanner variability of cortical overlap metrics in the SIMON dataset. Box-plots show Dice and Surface Dice between repeated scans, grouped by hemisphere. Overlap remains generally high, but reliability is not uniform across cortical regions or hemispheres, which motivates region-aware QC rather than assuming all structures are equally stable.

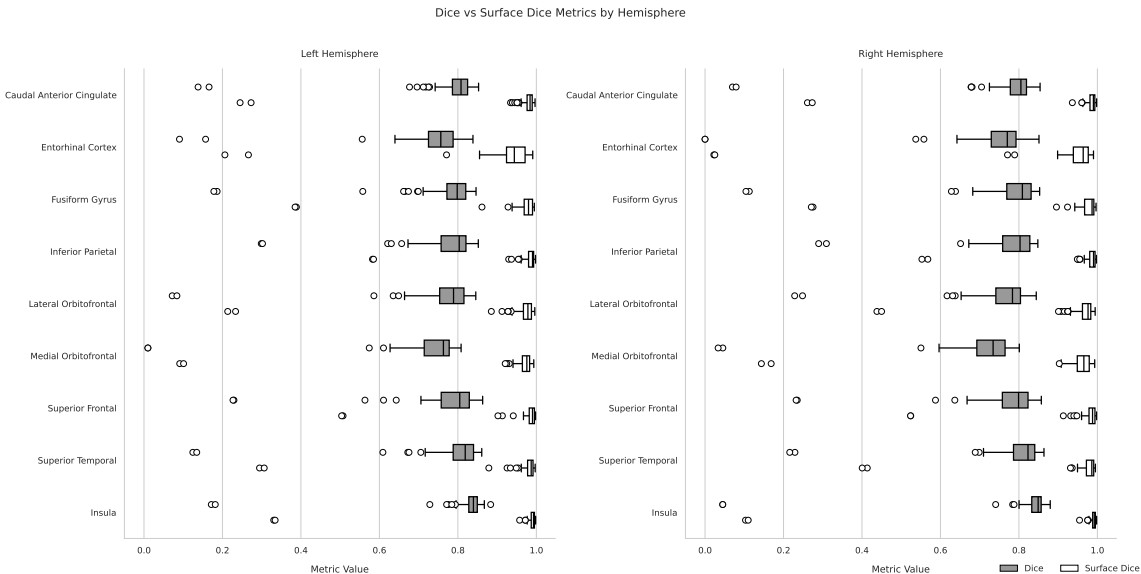

Figure 8: Dice and Surface Dice distributions across cortical regions in the left and right hemispheres of the SIMON dataset using FreeSurfer. Each structure is evaluated over multiple longitudinal scans from the same individual. Surface Dice (white boxes) consistently exceeds traditional Dice (gray boxes), especially in regions with complex geometry such as the entorhinal cortex and insula. A surface-aware criterion is less punitive than standard Dice and better separates minor boundary jitter from clear segmentation failures, making it a more informative QC signal for cortical morphometry.

