# OpenReview forum: "Benchmarking the Reproducibility of Brain Tissue Segmentation Across MRI Scanners"
_MIDL.io/2026/Validation_Papers — MIDL 2026 - Validation Papers Poster_

### Official Review · Reviewer_Enh9 · 2025-12-20

**Confidence:** 5
**Preliminary Rating:** 5

**Summary:**

This paper presents a rigorous evaluation of the reproducibility of brain tissue segmentation across multiple MRI scanners and longitudinal datasets. The authors benchmark two deep learning-based segmentation pipelines, FastSurfer and SynthSeg, against the FreeSurfer recon-all reference using two complementary datasets: a 17-year single-subject longitudinal dataset (SIMON) and a nine-site traveling-subject dataset (SRPBS). Segmentation reproducibility is quantified using Dice, Surface Dice, HD95, and MAPE, providing a detailed assessment of both volumetric and boundary-level agreement. Results reveal substantial inter-scanner variability, particularly in small subcortical structures (e.g., amygdala, pallidum), highlighting challenges in detecting subtle longitudinal changes. The study also explores registration, interpolation, and surface-based quality filtering, offering practical insights for harmonization and robust morphometry in multi-site neuroimaging studies.

**Strengths:**

Comprehensive benchmarking: The study systematically evaluates state-of-the-art pipelines across longitudinal and multi-site datasets, addressing a critical gap in reproducibility research.

Use of multiple metrics: By combining volumetric (MAPE) and spatial (Dice, Surface Dice, HD95) metrics, the paper provides a nuanced understanding of segmentation variability.

Longitudinal relevance: The inclusion of a 17-year single-subject dataset allows realistic assessment of method stability over extended periods, which is highly valuable for personalized brain health monitoring.

Practical guidance: Insights on registration strategies, interpolation modes, and quality-based filtering provide actionable recommendations for improving robustness in real-world pipelines.

Reproducibility and openness: The authors release code and workflows, supporting transparency and future method comparisons.

**Weaknesses:**

Lack of ground-truth labels: Both datasets are unlabeled, so the study evaluates reproducibility rather than true accuracy. While justified, this limits the ability to benchmark absolute segmentation performance.

Limited preprocessing exploration: The analysis does not examine the effect of denoising or intensity normalization, which may further influence reproducibility.

Focus on specific tools: While FreeSurfer, FastSurfer, and SynthSeg are widely used, including other high-performing segmentation models (e.g., nnU-Net, nnFormer) could strengthen generalizability.

Small sample for inter-scanner variability: The SRPBS dataset uses only nine subjects, which may limit statistical power for some conclusions about scanner effects.

Limited discussion on clinical impact: The paper could expand on how these reproducibility findings might influence longitudinal studies or clinical decision-making.

**Detailed Comments:**

Figures and tables are clear; consider adding effect sizes or confidence intervals for inter-scanner variability metrics.

Minor typos in metric definitions (e.g., spacing in formulas) could be cleaned up.

Discuss potential strategies for harmonization across scanners beyond quality filtering.

**Justification Of The Preliminary Rating:**

This paper provides a timely and rigorous evaluation of brain segmentation reproducibility, addressing a key bottleneck in multi-site and longitudinal neuroimaging research. The methodology is sound, results are clearly presented, and the work is highly relevant to both the machine learning and neuroimaging communities. While ground-truth labels are lacking, the focus on reproducibility is appropriate, and the inclusion of practical recommendations for harmonization and quality filtering adds strong applied value. Overall, the paper advances understanding of methodological variability in widely used segmentation pipelines and offers a resource for future benchmarking efforts.

**Questions To Address In The Rebuttal:**

Can the authors clarify how preprocessing (denoising, normalization) might influence reproducibility metrics?

How would results change if other segmentation models (e.g., nnU-Net, nnFormer) were included?

Could the authors provide guidance on acceptable thresholds for Surface Dice filtering in clinical studies?

---

### Official Review · Reviewer_YrKv · 2026-01-09

**Confidence:** 3
**Preliminary Rating:** 2
**Final Rating:** 4

**Summary:**

This paper presents a comprehensive validation study benchmarking the reproducibility of brain tissue segmentation across MRI scanners using two widely adopted deep learning–based pipelines, FastSurfer and SynthSeg integrated into FreeSurfer. The authors address the critical problem of scanner- and site-induced variability in longitudinal and multi-site brain morphometry, which limits the detection of subtle neuroanatomical changes relevant for clinical and translational studies. Using a 17-year single-subject longitudinal dataset and a multi-site traveling-subject dataset, the study systematically quantifies variability via complementary overlap-, surface-, distance-, and volume-based metrics. The work highlights that even state-of-the-art tools exhibit non-negligible variability—particularly in small subcortical structures—raising important concerns about sensitivity limits in individualized longitudinal analyses.

**Strengths:**

1. The combination of a 17-year single-subject longitudinal cohort and a multi-site traveling-subject dataset provides a rigorous and insightful testbed for reproducibility analysis.
2. Quantifying variability in small structures (e.g., amygdala, ventral diencephalon) directly informs the limits of detecting subtle longitudinal changes in real-world studies.
3. The inclusion of Dice, Surface Dice, HD95, and MAPE offers a multi-faceted view of segmentation consistency beyond a single metric.

**Weaknesses:**

1. The inclusion of Dice, Surface Dice, HD95, and MAPE offers a multi-faceted view of segmentation consistency beyond a single metric.
2. While the SIMON dataset is unique, conclusions about long-term trends and biases may not generalize beyond this individual.
3. Using FreeSurfer recon-all outputs as a reference while simultaneously evaluating FreeSurfer-derived methods introduces potential circularity and bias.
4. Variability is mostly described descriptively; formal statistical tests or variance decomposition (e.g., scanner vs. session effects) are limited.

**Detailed Comments:**

1. Clarify more explicitly why FreeSurfer recon-all is treated as the reference standard, given known biases and variability in FreeSurfer itself.
2. Consider adding confidence intervals or bootstrapped uncertainty estimates for key reproducibility metrics to better quantify robustness.

**Justification Of Final Rating:**

The authors have made a commendable effort to address the concerns raised in the initial review. In particular, they clarify that FreeSurfer recon-all is used as a common label space rather than a ground-truth reference, appropriately reframing the study as an evaluation of reproducibility rather than accuracy. The addition of formal statistical testing with effect sizes substantially strengthens the evidence that cross-scanner variability systematically degrades segmentation consistency, moving beyond largely descriptive analysis, and the interpretation of distance-based metrics is improved. Important limitations remain, including dependence on FreeSurfer-based label definitions, the absence of voxel-wise ground truth, and limited generalizability of long-term trends due to reliance on a single subject; however, these are now explicitly acknowledged and appropriately positioned as limitations and future directions. Overall, the revisions significantly improve rigor and clarity, supporting acceptance despite remaining structural constraints.

**Justification Of The Preliminary Rating:**

The methodology is sound and the experiments are carefully designed, clearly demonstrating that scanner-induced variability remains a major challenge for longitudinal brain morphometry. However, the lack of ground-truth annotations and the strong reliance on a single long-term subject limit the generalizability and strength of the conclusions.

**Questions To Address In The Rebuttal:**

1. How sensitive are the main conclusions to the choice of FreeSurfer recon-all as the reference standard?
2. Can the authors provide additional justification or evidence that observed volume trends are methodological rather than biological?
3. Do the authors expect similar long-term variability trends if multiple subjects with long longitudinal follow-up were available?

---

### Official Review · Reviewer_EmYH · 2026-01-09

**Confidence:** 4
**Preliminary Rating:** 4
**Final Rating:** 5

**Summary:**

The authors establish a reproducible benchmark to evaluate the robustness of state-of-the-art neuroimaging segmentation pipelines across scanners, sites, and time in multi-site and longitudinal MRI studies. They compare three widely used segmentation methods (FreeSurfer, FastSurfer, and SynthSeg) on two complementary multi-scan datasets (n=73 and n=411), using four metrics (Dice, Surface Dice, 95th percentile Hausdorff distance, and mean absolute percentage error) to quantify segmentation variability. The results show substantial segmentation variability associated with scanner and site differences. Overall, the study highlights the clinical need for quality-control strategies to enable robust neuroimaging analyses.

**Strengths:**

- **Clinical Relevance**: The paper introduces a benchmark for assessing brain MRI segmentation robustness in multi-site, multi-scanner, and longitudinal settings, which closely reflects real-world neuroimaging study conditions and practical research needs.
- **Community Value**: By systematically quantifying sensitivity of widely used segmentation pipelines to scanner/protocol/site variation, the work provides actionable evidence that can inform quality control, harmonization efforts, and study design in neuroimaging.
- **Comprehensive Evaluation**: The evaluation is thorough and well structured, leveraging two complementary public datasets with repeated scans (n=73 and n=411), three widely used segmentation pipelines (FreeSurfer, FastSurfer, SynthSeg), and multiple variability metrics (Dice, Surface Dice, 95th percentile Hausdorff distance, and mean absolute percentage error), providing a solid validation setup for the paper’s goals.
- **Reproducibility support**: The authors provide open-source code.

**Weaknesses:**

- The manuscript would benefit from at least one representative figure to help readers visually understand the MRI data and segmentation target. A qualitative figure comparing the same image/tissues across segmentation modules corresponding to Table 1 or across time corresponding to Tables 2 and 3, would be especially helpful. In addition, a small schematic figure to illustrate workflow/pipeline on high-level (data → segmentation → registration/interpolation → ROI analysis/metrics) would also improve readability. These additions could be placed in an expanded appendix.
- Since ground-truth labels are not available, the study primarily measures consistency/variability rather than absolute accuracy. It would strengthen the paper to better justify the key assumption of “relative stability of anatomy in healthy subjects” by citing supporting prior work (using same practice) from literature.
- To promote reproducibility, the authors omit data preprocessing steps (e.g., intensity normalization). It would strengthen the paper to justify this choice, either by citing prior work showing minimal impact on these segmentation pipelines or by adding a brief sensitivity analysis comparing results with and without normalization.
- Runtime is a practical limitation (~2 hours per subject on CPU). If GPU execution was not feasible in the current setup, providing an approximate estimate of expected GPU speedups would improve the paper’s translational relevance. Including memory requirements of segmentation methods along with inference times would help readers assess scalability.

**Detailed Comments:**

- Please include the number of parameters (model capacity) for each segmentation module to give idea of model size, along with estimated GPU processing time.
- It would be helpful to comment on other potential applications of this benchmark beyond brain MRI segmentation, and what adaptations would be needed for those settings.

**Justification Of Final Rating:**

The authors’ revision addressed my main concerns about presentation and computational details. They added a representative qualitative figure  (Fig. 1) and included a high-level schematic (Fig. 2). They also clarified that the study evaluates reproducibility/consistency rather than absolute accuracy and better justified the anatomical stability assumption with supporting prior work. Finally, they added model capacity details and representative GPU inference time information. Overall, the work provides a clinically relevant, well-designed, and reproducible benchmark with clear community value.

**Justification Of The Preliminary Rating:**

The paper has sound validation-related strengths such as clear clinical relevance, strong community value, comprehensive evaluation and reproducibility support. However, it lacks some details such as presentation (figures) and computational details.

**Questions To Address In The Rebuttal:**

Address weaknesses:
- The manuscript would benefit from at least one representative MRI figure to help readers visually understand the data and segmentation target. A qualitative figure comparing the same image/tissues across segmentation modules corresponding to Table 1 or across time corresponding to Tables 2 and 3, would be especially helpful.
- Draw a small schematic figure to illustrate workflow/pipeline on high-level: data → segmentation → registration/interpolation → ROI analysis/metrics.
- It would strengthen the paper to better justify the key assumption of “relative stability of anatomy in healthy subjects” by citing supporting prior work (using same practice) from literature.
- Cite prior work showing minimal impact of preprocessing on these segmentation pipelines or by adding a brief sensitivity analysis comparing results with and without normalization.
- Include key processing details such as memory requirements of segmentation methods along with inference times.

---

### Author Rebuttal · Authors · 2026-01-25

**Rebuttal:**

## Revision summary

* **Reframed scope**: removed any “reference standard / ground truth” wording; clarified we evaluate **reproducibility/consistency**, and FreeSurfer recon-all is a **reference label space / common baseline**, not accuracy GT.
* **Added visuals**: new **Figure 1** (qualitative SRPBS example with scanner/protocol appearance differences + contour overlays + Dice) and a **workflow schematic** (Figure 2).
* **Made assumptions explicit**: added test–retest justification for negligible anatomical change over short intervals (healthy subjects) + citations.
* **Preprocessing clarified**: stated we **avoid external denoising/normalization** beyond pipeline defaults to prevent study-specific choices; added this as a limitation/sensitivity axis (incl. note on generative harmonization directions).
* **Compute/model transparency**: added GCP setup (**64 vCPUs, 512 GB RAM**), FreeSurfer CPU runtime (**~2 h/subject**), and documented GPU infeasibility in our setup; included model sizes (SynthSeg **~20–40M** params; FastSurfer **~1.8–1.85M** params) + reference GPU runtimes.
* **Stronger statistics**: added Mann–Whitney U + BH-FDR showing cross-scanner vs within-scanner differences significant in **64/72 (89%)** ROI–metric pairs ((p_{\text{adj}}<0.05)), with large effects in **56/72** (Cliff’s (\delta>0.474)); added concrete examples (e.g., caudate Dice **0.868 vs 0.802**, putamen **0.897 vs 0.848**, hippocampus HD95 **1.234 vs 1.830 mm**).
* **HD95 made interpretable**: explained voxel-scale behavior (SIMON median **1.0 mm**, IQR=0 for **16/18** regions) vs SRPBS higher variability and **7–9 mm** site-pair outliers.
* **QC guidance added**: Appendix recommendations for **structure-specific Surface Dice thresholds** (5th percentile range **0.50–0.93**); propose **10th percentile** as “flag-for-review” cutoff; full threshold table in repo.
* **Limitations expanded**: explicitly note **small SRPBS N=9** and single-subject SIMON; outline future “anchor sensitivity” via re-anchoring (SPM/CAT/MUSE/NeuroQuant/consensus) and optional small manual subset.
* **Minor edits**: cleaned typos/spacing in metric formulas.

**Supporting Material:**

/attachment/2911a4cb159075f20968a2573e5913b0ec73d98a.pdf

---

### Meta-Review · Area_Chair_971h · 2026-02-09

**Recommendation:** Accept (Oral)
**Confidence:** 5

**Metareview:**

The paper had initially already received very positive remarks and undergone a very thorough revision, rebuttal and discussion phase. All reviewers praised the clarity, sound experiments and impactful findings including two strong accept votes. Only minor criticism remained regarding the reliance on FreeSurfer labels that is, however, now more clearly acknowledged. In summary the reviewers found that the submission excels amongst others with clinically relevant, well-designed and reproducible experiments that address a key bottleneck in multi-site and longitudinal neuroimaging research. I recommend acceptance as oral.

---

### Decision · Program_Chairs · 2026-02-14

Accept (Poster)